# Impact of Seminal Plasma Antioxidants on DNA Fragmentation and Lipid Peroxidation of Frozen–Thawed Horse Sperm

**DOI:** 10.3390/antiox13030322

**Published:** 2024-03-06

**Authors:** Jaime Catalán, Iván Yánez-Ortiz, Marc Torres-Garrido, Jordi Ribas-Maynou, Marc Llavanera, Isabel Barranco, Marc Yeste, Jordi Miró

**Affiliations:** 1Biotechnology of Animal and Human Reproduction (TechnoSperm), Institute of Food and Agricultural Technology, University of Girona, ES-17003 Girona, Spain; dr.jcatalan@gmail.com (J.C.); ivan.yanez22@gmail.com (I.Y.-O.); marctorres045@gmail.com (M.T.-G.); jordi.ribasmaynou@udg.edu (J.R.-M.); marc.llavanera@udg.edu (M.L.); 2Unit of Cell Biology, Department of Biology, Faculty of Sciences, University of Girona, ES-17003 Girona, Spain; 3Equine Reproduction Service, Department of Animal Medicine and Surgery, Faculty of Veterinary Sciences, Autonomous University of Barcelona, ES-08193 Cerdanyola del Vallès, Spain; 4School of Veterinary Medicine, Faculty of Medical, Health and Life Sciences, International University of Ecuador, Quito 170411, Ecuador; 5Department of Animal Medicine and Surgery, Faculty of Veterinary Medicine, University of Murcia, ES-30100 Murcia, Spain; isabel.barranco@um.es; 6Catalan Institution for Research and Advanced Studies (ICREA), ES-08010 Barcelona, Spain

**Keywords:** horse, semen, seminal plasma, sperm, cryopreservation, antioxidant, DNA fragmentation, lipid peroxidation

## Abstract

Cryopreservation is a stressful process for sperm, as it is associated with an increased production of reactive oxygen species (ROS). Elevated ROS levels, which create an imbalance with antioxidant capacity, may result in membrane lipid peroxidation (LPO), protein damage and DNA fragmentation. This study aimed to determine whether the membrane LPO and DNA fragmentation of frozen–thawed horse sperm relies upon antioxidant activity, including enzymes (superoxide dismutase (SOD), glutathione peroxidase (GPX), catalase (CAT) and paraoxonase type 1 (PON1)); non-enzymatic antioxidant capacity (Trolox-equivalent antioxidant capacity (TEAC), plasma ferric reducing antioxidant capacity (FRAP) and cupric reducing antioxidant capacity (CUPRAC)); and the oxidative stress index (OSI) of their seminal plasma (SP). Based on total motility and plasma membrane integrity (SYBR14^+^/PI^−^) after thawing, ejaculates were hierarchically (*p* < 0.001) clustered into two groups of good- (GFEs) and poor-(PFEs) freezability ejaculates. LPO and DNA fragmentation (global DNA breaks) were higher (*p* < 0.05) in the PFE group than in the GFE group, with LPO and DNA fragmentation (global DNA breaks) after thawing showing a positive relationship (*p* < 0.05) with SP OSI levels and ROS production. In addition, sperm motility and membrane integrity after thawing were negatively (*p* < 0.05) correlated with the activity levels of SP antioxidants (PON1 and TEAC). The present results indicate that LPO and DNA fragmentation in frozen–thawed horse sperm vary between ejaculates. These differences could result from variations in the activity of antioxidants (PON1 and TEAC) and the balance between the oxidant and antioxidant components present in the SP.

## 1. Introduction

Sperm cryopreservation is a fundamental technique for assisted reproduction in mammals due to the benefits that its use offers [1], such as preserving genetic material for an unlimited time and facilitating its availability and transport [2]. The current protocols of freezing and thawing may, however, reduce the survival and fertilization capacity of mammalian sperm [3]. In horses, one of the consequences of conducting artificial insemination (AI) with frozen–thawed sperm is a reduced pregnancy rate compared to fresh or liquid-stored semen [4,5,6]. This lower performance can be explained by the detrimental effect of cryopreservation on the integrity of the sperm plasma membrane, acrosome and DNA, as well as on sperm motility and mitochondrial activity [7,8,9,10].

Sperm are damaged by cryopreservation because it induces cryogenic and osmotic stress, which in turn increases the rate of reactive oxygen species (ROS) production [3,11] and alters cellular antioxidant defense systems [11]. When ROS levels exceed a cell’s inherent physiological antioxidant defenses, the establishment of a condition of oxidative stress (OS) leads to cellular damage [12,13,14]. Indeed, some of the distinctive characteristics of sperm make them more susceptible to ROS damage: the high content of polyunsaturated fatty acids (PUFAs) in their membranes, their large number of mitochondria and the reduced content of antioxidants in their cytoplasm [15]. The dynamics of the sperm plasma membrane play an important role in the processes of maturation, capacitation and fertilization [16]. An increase in OS can damage the plasma membrane and cause lipid peroxidation (LPO), thus modifying its fluidity and permeability and ultimately resulting in cell death [17]. Another frequent effect of OS is sperm DNA damage. The integrity of sperm DNA is crucial for successful fertilization, subsequent embryonic development and implantation and the maintenance of a healthy pregnancy [18,19]. Sperm DNA is characterized by the great condensation and structural organization of the chromatin present in the nucleus, which results from the replacement of histones by transition proteins and later by protamines during spermiogenesis [9]. This organization provides protection for the genetic material against oxidative damage. When the condensation process is altered due to incomplete protamination, sperm DNA becomes an easy target for oxygen free radicals [20]. Specifically, OS can cause DNA fragmentation and affect one (single-strand breaks, SSBs) or both (double-strand breaks, DSBs) strands of the DNA [21].

In raw semen, due to the low antioxidant capacity of their spermatozoa, enzymatic and non-enzymatic antioxidants present in seminal plasma (SP) are the main line of defense for sperm against the damage induced by OS [22,23,24,25]. However, whether the effects of SP during the storage of chilled and cryopreserved equine semen are beneficial or detrimental is not clear [3,26,27,28,29,30]. For this reason, most SP is routinely removed before horse sperm cryopreservation, as some SP proteins have a negative effect on sperm motility and viability during storage [29,30]. Yet, the removal of SP increases the degree of sperm susceptibility to oxidative damage, as the enzymatic and non-enzymatic antioxidants that this fluid contains are also removed [29,31,32]. Moreover, in horses, as in other mammalian species, there is great variability between, and even within, stallions in the ability of their sperm to withstand cryopreservation [27,33,34,35,36,37], leading to the classification of ejaculates/stallions as having good freezability (GFE) or poor freezability (PFE) [28,30,35,38]. This variability could be related to differences in the composition of SPs and the activity exerted by their antioxidants [39,40,41]. In this regard, a recent study by our research group focused on the relationship between the levels of some antioxidants in SP and the resilience of horse sperm to freezing and thawing [29]. To the best of our knowledge, however, whether the DNA integrity and LPO of the plasma membrane of frozen–thawed horse sperm rely upon the activity of enzymatic and non-enzymatic SP antioxidants and the OS index (OSI) has not been investigated. The present study, therefore, aimed to determine if the antioxidant components of horse SP, including enzymatic (superoxide dismutase (SOD), glutathione peroxidase (GPX), catalase (CAT) and paraoxonase type 1 (PON1)) and non-enzymatic antioxidants (measured in terms of Trolox-equivalent antioxidant capacity (TEAC), the ferric-reducing ability of plasma (FRAP), the copper-reducing antioxidant capacity (CUPRAC)) and the OSI, are related to the DNA fragmentation (SSBs and/or DSBs) and membrane LPO of frozen–thawed horse sperm.

## 2. Materials and Methods

### 2.1. Reagents and Suppliers

Unless otherwise stated, all chemicals and reagents were purchased from Merck (Merck KgaA, Darmstadt, Germany). 

### 2.2. Animals and Samples

Twenty ejaculates from different stallions (between 5 and 15 years old) of several breeds (Andalusian, Warmblood and Arabian), mature and with proven fertility, were collected. Their diet included mixed hay and basic concentrate, without antioxidant supplementation; water was provided ad libitum. All stallions were clinically healthy and housed in individual stalls at the Equine Reproduction Service, Autonomous University of Barcelona (Bellaterra, Cerdanyola del Vallès, Spain). This is a center approved by Generalitat de Catalunya (Regional Government of Catalonia), Spain, and the European Union (authorization number: ES09RS01E) for the collection of equine semen, which operates under rigorous health and animal welfare protocols. As no manipulation of animals was carried out beyond semen collection, the Ethics Committee of the Autonomous University of Barcelona indicated that no additional ethical approval was necessary to carry out this study. On the other hand, the sanitary guidelines established by the Council of the European Communities in Directive 82/894/EEC of 21 December 1982 were complied with, as stallions were free from equine infectious anemia, equine contagious metritis and equine viral arteritis.

Semen was collected using an artificial vagina (Hannover model; Minitüb GmbH, Tiefenbach, Germany) previously heated to 48 °C–50 °C, coupled with a nylon filter to remove the gel fraction. After obtaining the ejaculate, the gel fraction was excluded, and a 10 µL sample was taken to assess the sperm concentration using a Neubauer chamber (Paul Marienfeld GmbH & Co. KG, Lauda-Königshofen, Germany). Next, each ejaculate was divided into two aliquots: the first was used to obtain SP by centrifugation, and the second was diluted 1:5 (*v*:*v*) with a Kenney extender [42], used for sperm quality analysis (sperm motility, morphology and viability) and then cryopreserved. While sperm motility was assessed using a computer-aided sperm analysis (CASA) system (as detailed in Section 2.3), sperm viability and morphology were examined through eosin–nigrosine staining [43]. All semen samples included in this study adhered to established sperm quality criteria prior to freezing, with thresholds of 65% for total motility, 65% for viable sperm and 70% for morphologically normal sperm.

### 2.3. Isolation of SP

Immediately after collection and in order to isolate the SP, ejaculates were centrifuged at least five times at 1500× *g* and 4 °C for 10 min (Medifriger BL-S; JP Selecta S.A., Barcelona, Spain), as it has been previously described [29,44,45]. After each centrifugation cycle was completed, the supernatant was carefully examined to ensure the absence of sperm under a phase contrast microscope (Olympus Europa, Hamburg, Germany). The required number of centrifugations depended on each ejaculate, and thus samples were centrifuged as many times as necessary until they were free from sperm. Finally, SP samples were stored at −80 °C until the time of analysis. When all SP samples were collected, they were thawed on ice to measure the antioxidant activity.

### 2.4. Evaluation of Antioxidant Activity Levels of SP

#### 2.4.1. Enzymatic Antioxidants

The enzymatic antioxidants evaluated in horse SP were SOD, CAT, GPX and PON1 [29,45]. The levels of activity of SOD, CAT and GPX were measured using commercially available assays, following the manufacturer’s instructions (SOD and GPX: Randox, Crumlin, United Kingdom; CAT: Sigma-Aldrich, St. Louis, MO, USA). In brief, the SOD activity (Ransod kit) relied on the creation of a red formazan dye produced through the interaction of 2-(4-iodophenyl)-3-(4-nitrophenol)-5-phenyltetrazolium chloride (INT) with a superoxide radical generated by xanthine and xanthine oxidase. The evaluation of GPX activity (Ransel kit) involved the oxidation of the reduced form of nicotinamide adenine dinucleotide phosphate (NADPH) using cumene hydroperoxide and glutathione reductase. Finally, the assay for assessing the antioxidant catalytic activity of CAT relied on the inhibition of color development in a urate assay that produces H_2_O_2_, wherein urate is present in excess. H_2_O_2_ production was quantified using the Trinder (4-aminophenazone and 3,5-dichloro-2-hydroxybenzenesulphonate) reagent along with horseradish peroxidase. 

To measure the activity levels of PON1, the protocol described by Barranco et al. [46] was followed and adapted to the SP of horses, specifically measuring the hydrolysis of 4-nitrophenyl acetate into 4-nitrophenol. While the determination of PON1, SOD and GPX activity was performed with an Olympus AU400 chemistry analyzer (Olympus Europe GmbH, Hamburg, Germany), CAT activity was measured using a microplate reader (PowerWave XS; Bio-Tek Instruments, Winooski, VT, USA). The levels of PON1 and GPX were expressed in IU/L, whereas those of SOD and CAT were expressed in IU/mL.

#### 2.4.2. Non-Enzymatic Antioxidants

The activity levels of non-enzymatic antioxidants were analyzed on the basis of CUPRAC, FRAP and TEAC, following the protocols described by Li et al. [47] adapted to horse SP. The CUPRAC assay evaluates the antioxidant capacity of a sample by measuring its ability to convert Cu^2+^ to Cu^1+^ in the presence of a chelating agent that forms stable colored complexes with Cu^1+^ [48,49]. The FRAP assay operates on the principle of reducing the ferrictripyridyltriazine (Fe^3+^-TPTZ) complex to ferrous tripyridyltriazine (Fe^2+^-TPTZ) at low pH through the action of antioxidants in the sample. The resulting blue-colored product (Fe^2+^-TPTZ) undergoes changes in absorbance that are correlated to the antioxidant capacity of the sample [50,51]. Lastly, the TEAC assay measures a sample’s ability to reduce or inhibit the formation of oxidized products generated in the assay, such as the 2,2′-azino-bis (3-ethylbenzthiazoline-6-sulfonic acid) radical cation (ABTS^•+^), a blue-green chromophore that decreases in intensity in the presence of antioxidants [51,52]. For all the above determinations, an Olympus AU400 automated chemistry analyzer (Olympus Europe GmbH) was used. The activity levels of CUPRAC, FRAP and TEAC were expressed as mmol Trolox equivalent/L. In each test, the coefficient of variation—both for enzymatic and non-enzymatic antioxidants—was less than 10%.

### 2.5. Determination of the Oxidative Stress Index (OSI)

The OSI of SP was determined using the following calculation: OSI (arbitrary unit) = total oxidative status (TOS, μmol H_2_O_2_ equivalent/L)/TEAC (mmol Trolox equivalent/L) [53]. To calculate the TOS as described by Erel. [54], a protocol previously adjusted to horse semen was used. This test relies on the conversion of Fe^2+^ to Fe^3+^ in the presence of oxidants in an acid medium, followed by the quantification of Fe^3+^ using xylenol orange. The assessment was conducted using an Olympus AU400 automated chemistry analyzer. TOS outcomes were expressed in terms of μmol H_2_O_2_ equivalent/L.

### 2.6. Sperm Cryopreservation

Before the cryopreservation procedure, samples underwent centrifugation at 660× *g* and 20 °C for 15 min. Subsequently, the supernatant was discarded, and the resulting pellet was re-suspended in a commercial freezing medium (BotuCRIO^®^; Botupharma Animal Biotechnology, Botucatu, Brazil) containing 1% glycerol and 4% methylformamide as permeable cryoprotectants. Following this, the sperm concentration and viability of each sample were assessed, and the same freezing medium was introduced to achieve a final concentration of 200 × 10^6^ viable sperm/mL. Finally, the samples were loaded into 0.5 mL straws and subjected to freezing using an automatic speed-controlled freezer (Ice-Cube 14S; Minitüb GmbH, Tiefenbach, Germany). The freezing process involved the following stages: (1) cooling from 20 °C to 5 °C for 60 min at a rate of 25 °C/min, (2) freezing between 5 °C and 90 °C for 20 min at a rate of 4.75 °C/min and the last phase of freezing between 90 °C and 120 °C for 2.7 min at a speed of 11.11 °C/min. Upon completion of this process, the straws were immersed in liquid nitrogen at −196 °C and stored in suitable tanks.

Samples were thawed in a circulating water bath previously heated to 38 °C for 30 s. Regarding the content of each of the samples, two straws were thawed per ejaculate. Its contents were pooled in a 10 mL conical tube and then diluted again (1:2, *v*/*v*) with the Kenney extender [42] preheated to 38 °C. The sperm quality and functionality parameters evaluated in each frozen–thawed sample were (1) the motility, (2) the integrity of the plasma membrane and (3) acrosome, (4) the disorder of membrane lipids, (5) the potential of the mitochondrial membrane (MMP), (6) the intracellular levels of ROS (general ROS and superoxides), (7) the membrane LPO and (8) the sperm DNA fragmentation (more detailed in Section 2.7, Section 2.8 and Section 2.9).

### 2.7. Assessment of Sperm Motility

Sperm motility was assessed utilizing a Computer-Assisted Sperm Analysis (CASA) system (ISAS v. 1.0.; Proiser S.L.; Valencia, Spain), as it has been previously described [29]. In brief, 5 μL of each semen sample (diluted in Kenney extender at a concentration of 50 × 10^6^ spermatozoa/mL) was applied to a preheated (38 °C) Makler chamber (Sefi Medical Instruments; Haifa, Israel). Subsequently, the samples were examined under a 10× negative phase contrast objective lens using an Olympus B×41 microscope (Olympus, Tokyo, Japan), equipped with a plate heated to 38 °C. A minimum of 1000 sperm cells were counted per analysis. Percentages of total motility (TM, %) and progressive motility (PM, %) were documented in each evaluation, along with kinetic parameters, including straight-line velocity (VSL, μm/s), curvilinear velocity (VCL, μm/s), average path velocity (VAP, μm/s), percentage of straightness (STR, %), percentage of linearity (LIN, %), percentage of oscillation (WOB, %), frequency of head displacement (BCF, Hz) and mean amplitude of lateral head displacement (ALH, μm). The CASA settings used adhered to the recommendations provided for horse sperm, which included the following: frames/s: 25 images captured per second; connectivity: 6; particle area: >4 and <75 μm^2^; and minimum number of images to calculate the ALH: 10. The cutoff value for motile sperm was VAP ≥ 10 μm/s, and for progressively motile sperm, it was STR ≥ 75%.

### 2.8. Sperm Functionality Analysis

Flow cytometry was employed to analyze sperm functionality parameters, which encompassed the acrosome integrity (*Arachis hypogaea* (peanut) agglutinin–fluorescein isothiocyanate (PNA-FITC)/propidium iodide (PI)), the plasma membrane integrity (SYBR14/PI), the intracellular levels of ROS (2,7-dichlorodihydrofluorescein and diacetate (H_2_DCFDA)/PI), the mitochondrial membrane potential (MMP; 5,5′,6,6′-tetrachloro-1,1′3,3′tetraethyl-benzimidazolylcarbocyanine iodide (JC-1)) and the superoxides anion (O_2_^−^) (hydroethidine (HE)/1-(4-[3-methyl-2,3-dihydro-(benzo-1,3-oxazole)-2-methylidene]-quinolinium)-3-trimethylammonium propane diodide (YO-PRO-1)), plasma membrane lipid disorder (Merocyanine 540 (M540)/YO-PRO-1) and LPO (BODIPY^581/591^-C11). The flow cytometer employed for the analysis was a CytoFLEX (Beckman Coulter, Fullerton, CA, USA) with a sheath flow rate set at 10 µL/min. Fluorochromes were obtained from Molecular Probes^®^ (Thermo Fisher Scientific, Waltham, MA, USA) and reconstituted in di-methyl sulfoxide (DMSO; Merck). The analyses adhered to the guidelines of the International Society for Advance Cytometry (ISAC) [55]. Before staining, the sperm concentration was standardized to 1 × 10^6^ sperm/mL. Each sample underwent analysis for a total of 10,000 events, and three technical replicates were assessed.

The samples were stimulated using a 488 nm argon ion laser with a power output of 50 mW. Two distinct dot plot distributions were utilized to eliminate (1) cellular aggregates, utilizing the dot plot distribution of forward scatter height (FSC-H) and altitude (FSC-A), and (2) cellular debris, based on the FSC-A distribution and side scatter altitude (SSC-A) dot plots. Four distinct optical filters were employed: (1) FITC with a band-pass filter of 525–540 nm for the analysis of PNA-FITC, SYBR14, JC-1 monomers (JC-1_mon_), dichlorofluorescein (DCF^+^), oxidized BODIPY^581/591^-C11 and (YO-PRO-1); (2) PE with a band-pass filter of 585–542 nm for the analysis of JC-1 aggregates (JC-1_agg_), BODIPY^581/591^-C11 not oxidized, and fluorescent ethidium (E^+^); (3) ECD with a band-pass filter of 610–620 nm for analyzing M540; and (4) PC5.5 with a band-pass filter of 690–650 nm for analyzing PI. The data recorded for each event (FSC-A, FSC-H, SSC-A, FITC, PE and PC5.5) were compiled into XIT files and processed using the CytExpert analysis software (CytExpert version 2.5, Beckman Coulter, Fullerton, CA, USA) to quantify sperm populations. The mean and standard error of the mean (SEM) were then calculated for each parameter.

#### 2.8.1. Assessment of Plasma Membrane Integrity (SYBR14/PI)

The sperm plasma membrane integrity was assessed using a LIVE/DEAD sperm viability kit (SYBR14/PI), following the procedure outlined by Garner and Johnson [56], with adaptations for horse sperm. In summary, semen samples were incubated with SYBR14 (final concentration: 31.8 nM) for 10 min. Following that, PI (final concentration: 7.6 µM) was applied at 38 °C in the dark for 5 min. Three discrete sperm populations were distinguished: (1) sperm with a damaged plasma membrane (SYBR14^+^/PI^+^), (2) sperm with a damaged plasma membrane (SYBR14^−^/PI^+^) and (3) sperm with an intact plasma membrane (SYBR14^+^/PI^−^). Particles devoid of staining (SYBR14^−^/PI^−^) were considered non-sperm debris and were employed for data correction in other analyses. A compensation of 8.34% was applied for SYBR14 spill over in the PC5.5 channel. 

#### 2.8.2. Assessment of Acrosome Integrity (PNA-FITC/PI)

The integrity of sperm acrosomes was evaluated using a combination of PNA-FITC and PI, following the procedure outlined by Rathi et al. [57]. Semen samples were subjected to incubation in the dark at 38 °C for 10 min, employing PNA conjugated with FITC (final concentration: 1.17 µg/mL) and PI (final concentration: 5.6 µM). This process led to the identification of four distinct sperm populations: (1) sperm with a damaged plasma membrane (PNA-FITC^+^/PI^−^), (2) sperm with a damaged plasma membrane accompanied by a fully lost outer acrosome membrane (PNA-FITC^−^/PI^+^), (3) sperm with a damaged plasma membrane displaying an outer acrosome membrane that was not completely intact (PNA-FITC^+^/PI^+^) and (4) viable sperm with an intact acrosome membrane (PNA-FITC^−^/PI^−^). Notably, no compensation was deemed necessary.

#### 2.8.3. Assessment of Mitochondrial Membrane Potential (JC-1)

The sperm’s mitochondrial membrane potential (MMP) was assessed using JC-1, following the procedure outlined by Ortega-Ferrusola et al. [58]. In a concise manner, semen samples were exposed to JC-1 (final concentration: 750 nM) at 38 °C in the dark for 30 min. JC-1 molecules exhibited orange-fluorescent aggregates (JC-1_agg_) in the presence of a high MMP while maintaining a green-fluorescent monomeric state (JC-1_mon_) in the presence of a low MMP. Two discernible sperm populations were identified: (1) sperm displaying a low mitochondrial membrane potential (MMP) (JC-1_mon_ fluorescence intensity higher than JC-1_agg_) and (2) sperm exhibiting a high MMP (JC-1_agg_ fluorescence intensity higher than JC-1_mon_). For each population, the fluorescence intensities of JC-1_agg_ and JC-1_mon_ were measured, and the ratio between them was subsequently calculated. No data compensation was applied.

#### 2.8.4. Intracellular Reactive Oxygen Species: Total ROS (H_2_DCFDA/PI) and O_2_^−^ (HE/YO-PRO-1)

The intracellular ROS levels of sperm were analyzed using oxidation-sensitive fluorescent probes: H_2_DCFDA to evaluate overall ROS and HE to measure superoxide anion (O_2_^−^) [59]. The discrimination between viable and non-viable spermatozoa was conducted using PI (for H_2_DCFDA) or YO-PRO-1 (for HE), according to the adapted protocol of Guthrie and Welch [60].

To measure overall ROS, semen samples were treated with H_2_DCFDA (final concentration: 50 µM) at 38 °C in the dark for 20 min, followed by incubation with PI (final concentration: 6 µM) for 5 min. When ROS are present, there is a de-esterification and oxidation of H_2_DCFDA to DCF^+^, a highly fluorescent molecule. Four sperm populations were distinguished: (1) non-viable sperm with low levels of ROS (DCF^−^/PI^+^), (2) viable sperm with low levels of ROS (DCF^−^/PI^−^), (3) non-viable sperm with high levels of ROS (DCF^+^/PI^+^) and (4) viable sperm with high levels of ROS (DCF^+^/PI^−^). The fluorescence intensity of DCF^+^ was quantified in all sperm populations, and no data compensation was applied.

For O_2_^−^ analysis, semen samples were exposed to HE (final concentration: 5 µM) and YO-PRO-1 (final concentration: 31.25 nM) for 30 min at 38 °C in the absence of light. In the presence of O_2_^−^, HE underwent oxidation from E^+^. Four different sperm populations were characterized: (1) non-viable sperm with low levels of O_2_^−^ (E^−^/YO-PRO-1^+^), (2) viable sperm with low levels of O_2_^−^ (E^−^/YO-PRO-1^−^), (3) non-viable sperm with high levels of O_2_^−^ (E^+^/YO-PRO-1^+^) and (4) viable sperm with high levels of O_2_^−^ (E^+^/YO-PRO-1^−^). The fluorescence intensity of E+ was measured in all sperm populations. A spill-over correction of E^+^ into the FITC channel (3.62%) was applied.

#### 2.8.5. Plasma Membrane Lipid Disorder (M540/YO-PRO-1)

The analysis of plasma membrane lipid disorder in sperm was conducted using the combination of M540/YO-PRO-1, following the protocol outlined by Rathi et al. [57], adapted to horse sperm with minor modifications [61]. In brief, semen samples were subjected to incubation in the absence of light at 38 °C for 10 min with M540 (final concentration: 2.5 µM) and YO-PRO-1 (final concentration: 25 nM). This process resulted in the identification of four distinct sperm populations: (1) non-viable sperm with high plasma membrane lipid disorder (M540^+^/YO-PRO-1^+^), (2) viable sperm with high plasma membrane lipid disorder (M540^+^/YO-PRO-1^−^), (3) non-viable sperm with low plasma membrane lipid disorder (M540^−^/YO-PRO-1^+^) and (4) viable sperm with low plasma membrane lipid disorder (M540^−^/YO-PRO-1^−^). No data compensation was required. 

#### 2.8.6. Membrane Lipid Peroxidation (LPO)

LPO was measured with a BODIPY^581/591^-C11 (BP) probe (Invitrogen Molecular Probes, Eugen, OR, USA). As egg yolk is known to bind to lipophilic BP [62], egg yolk containing the freezing medium (BotuCRIO^®^) was first removed by centrifugation through a density gradient (EquiPure™, Nidacon Laboratories AB, Gothenburg, Sweden) following the protocol described by the manufacturer. Subsequently, the BP probe was added to a 20 × 10^6^/mL suspension of post-thawed sperm at a final concentration of 2 μM. Next, sperm were incubated at 38 °C for 30 min, and then the samples were diluted 10 times in PBS 1× and analyzed through flow cytometry. The fluorochrome was excited with a blue laser (488 nm), and fluorescence was collected at ~590 nm (orange) and ~510 nm (green), depending on whether BP was intact or oxidized, respectively. The ratio of green to orange fluorescence (BP oxidized/BP intact) was used to measure the LPO of sperm. Positive controls were obtained after the addition of tert-butyl hydroperoxide (Luperox TBH70X, Sigma Aldrich, Saint Louis, MO, USA).

### 2.9. Evaluation of DNA Fragmentation

To assess the incidence of global DNA breaks (SSBs and DSBs) and DSBs in sperm DNA, alkaline and neutral Comet assays were performed, respectively, according to the protocol described by Casanovas et al. [63], adapted to horse sperm.

#### 2.9.1. Preparation of Semen–Agarose Plates and Lysis of Samples

The sperm concentration of the samples was adjusted to 1 × 10^6^ sperm/mL. Subsequently, samples were combined with 1% low-melting-point agarose (37 °C) in a ratio of 1:2 (*v*:*v*) to achieve an agarose concentration of 0.66%. Following this, 6.5 µL of the mixture was placed on a slide pre-treated with 1% agarose for gel adhesion and covered with a coverslip. The agarose–sperm mixture was allowed to gel at 4 °C for 5 min in contact with a cold metal plate on ice. Coverslips were then carefully removed. Following that, two slides were prepared—one for the alkaline comet assay and another for the neutral comet assay. Both slides were immersed initially in the first lysis solution, comprising 0.8 M of Tris-HCl, 0.8 M of DTT and 1% SDS (pH = 7.5), for a duration of 30 min. Subsequently, they were placed in the second lysis solution, which contained 0.4 M of Tris-HCl, 0.4 M of DTT, 50 mM of EDTA, 2 M of NaCl and 1% Tween20 (pH = 7.5), for an additional 30 min. Following these two incubations, the slides underwent a 2 min wash with distilled water. 

#### 2.9.2. Electrophoresis and Dehydration

Different procedures were employed for each comet assay variant, namely alkaline and neutral assays. In the case of the alkaline comet assay, slides were submerged in a chilled (4 °C) alkaline solution comprising 0.03 M of NaOH and 1 M of NaCl for a duration of 5 min, followed by electrophoresis in an alkaline buffer (0.03 M of NaOH; pH = 13) at 1 v/cm for 4 min. Conversely, for the neutral comet assay, slides underwent electrophoresis in a TBE buffer (0.445 M of Tris-HCl, 0.445 M of boric acid and 0.01 M of EDTA; pH = 8) at 1 v/cm for 12.5 min and were subsequently incubated in a 0.9% NaCl solution for 2.5 min. After electrophoresis, both slides underwent immersion in a neutralization solution with a composition of 0.4 M of Tris-HCl (pH = 7.5) for a duration of 5 min. Subsequently, they were subjected to dehydration through an ethanol series, including steps with concentrations of 70%, 90% and 100%, each lasting 2 min. The final step involved the horizontal drying of the slides.

#### 2.9.3. Staining and Imaging

The samples underwent staining through incubation with 1× SYTOX Orange (Invitrogen, Waltham, MA, USA) for 15 min at room temperature. Following this, slides were rinsed in distilled water for 2 min and left to dry horizontally. Subsequently, the stained samples were examined using a Zeiss Imager Z1 epifluorescence microscope (Carl Zeiss AG, Oberkochen, Germany) at a magnification of 100×. The capture of comets and tails was performed utilizing Axiovision 4.6 software (Carl Zeiss AG, Oberkochen, Germany), with care taken to avoid overexposure during capture.

#### 2.9.4. Comet Analysis

For the analysis of individual comets, the open access software Comet Score v2.0 (RexHoover) was utilized. Initially, the background of each image was adjusted, and individual comets were examined using the automatic analysis option. Subsequently, a manual review of the analysis was conducted to eliminate non-comet captures, address overlapping comets and correct comet head/tail detection. A minimum of 100 accurately analyzed cells/comets was required for each sample; if this threshold was not met, additional photographs were taken before repeating the process. The Comet Score software offers a range of parameters defining various aspects of the DNA within a given cell. Among these, the Olive Tail Moment (OTM) was selected as a quantitative measure for the incidence of DNA breaks, as various reports suggest it is the most informative parameter for this purpose [64,65]. The OTM is calculated as (tail mean intensity—head mean intensity) × %tail DNA/100. DNA breaks determined by the alkaline comet were considered to provide the global incidence of DNA breaks (SSBs + DSBs), and those given by the neutral comet were assumed to correspond to double-strand breaks (DSBs). 

### 2.10. Statistical Analyses

Figure preparation and data analysis were carried out using the GraphPad Prism software (Version 8.4.0, GraphPad Software LLC; San Diego, CA, USA) and R statistical package (Version 4.0.3, R Core Team; Vienna, Austria), respectively. First, data normality was checked through the Shapiro–Wilk test, and variance homogeneity was assessed using the Levene test. If necessary, an arcsine √x transformation was applied to achieve a normal distribution. For all analyses, the minimum level of statistical significance was established at *p* ≤ 0.05.

#### 2.10.1. Classification of Ejaculates Based on Their Cryotolerance

Ejaculates were categorized according to their cryotolerance (GFE and PFE) using the protocol specified by Morató et al. [66]. Post-thaw percentages of TM and sperm featuring an intact plasma membrane (sperm viability, SYBR14^+^/PI^−^) were documented for each sample, facilitating the execution of a whole-ligament hierarchical cluster analysis across the entire dataset using Euclidean distances. The results presented in the text are conveyed as means ± SEM.

#### 2.10.2. Comparison of LPO and DNA Fragmentation between Good- (GFEs) and Poor-Freezability Ejaculates (PFEs)

The comparison of the results for LPO and DNA fragmentation (incidence of global and double-strand DNA breaks) between GFEs and PFEs was conducted using a *t*-test for independent samples. Prior to the analysis, it was confirmed that the data distribution was normal and the variances were homogeneous. In cases where, despite data transformation, parametric assumptions were not met, the Mann–Whitney test was employed as an alternative. Results presented in the text are expressed as means ± SEM.

#### 2.10.3. Correlations of Membrane LPO and DNA Fragmentation with Post-Thaw Sperm Quality Parameters and Activity Levels of Enzymatic and Non-Enzymatic Antioxidants and the OSI of SP

The relationship of membrane LPO and sperm DNA fragmentation (incidence of global and double-strand DNA breaks) with sperm functionality (SYBR14^+^/PI^−^, JC-1_agg_, DCF^+^/PI^−^, E^+^/YO-PRO-1^−^, PNA-FITC^−^/PI and M540^+^/YO-PRO-1^−^), motility parameters (TM, PM, VSL, VCL, VAP, STR, LIN, WOB, ALH and BCF) and activity levels of enzymatic (GPX, CAT, SOD and PON1) and non-enzymatic (TEAC, CUPRAC and FRAP) antioxidants in SP and the SP OSI was assessed by computing Pearson’s correlation coefficients. In cases where, even after data transformation, parametric assumptions were not fulfilled, the Spearman correlation test was employed as an alternative.

## 3. Results

### 3.1. Classification of Horse Ejaculates into GFE and PFE Groups According to Their Cryotolerance

A hierarchical clustering analysis (*p* < 0.001) based on post-thaw TM and sperm viability (SYBR14^+^/PI^−^) resulted in the classification of the 20 horse ejaculates into 12 categorized as GFEs and 8 as PFEs. As illustrated in Figure 1, the ejaculates designated as GFEs exhibited significantly higher values (*p* < 0.001) for both TM and sperm viability (SYBR14^+^/PI^−^) compared to those classified as PFEs (66.18 ± 2.81% vs. 35.54 ± 4.36% and 67.69 ± 1.66% vs. 42.03 ± 2.49%, respectively).

### 3.2. Lipid Membrane Peroxidation

The sperm plasma membrane LPO (Figure 2) values were significantly higher (*p* < 0.001) in the ejaculates classified as PFEs compared to those classified as GFEs (19.03 ± 0.54 vs. 24.48 ± 1.12, respectively).

### 3.3. DNA Fragmentation

The global incidence of DNA damage (Figure 3a; alkaline comet) in the ejaculates classified as PFEs was significantly higher (*p* < 0.0001) than in those classified as GFEs (21.51 ± 0.87 vs. 15.99 ± 0.70, respectively). No significant differences (*p* = 0.069) in the incidence of double-strand DNA breaks (Figure 3b; neutral comet) were, however, observed between the PFEs and GFEs (1.17 ± 0.03 vs. 1.10 ± 0.02, respectively).

### 3.4. Correlations of the Degree of LPO and Sperm DNA Fragmentation with Sperm Motility Parameters after Thawing

Figure 4 shows the correlations between the membrane LPO and sperm DNA fragmentation (the incidence of global and double-strand DNA breaks) with the motility parameters of the post-thaw horse sperm. The sperm membrane LPO was negatively correlated with the TM (r = −0.7335, *p* = 0.0002), the PM (r = −0.5678, *p* = 0.0090), the VCL (r = −0.6104, *p* = 0.0043), the VSL (r = −0.4987, *p* = 0.0252), the VAP (r = −0.6196, *p* = 0.0036) and the ALH (r = −0.5850, *p* = 0.0067). The incidence of global sperm DNA damage (alkaline comet) was negatively related to the TM (r = −0.6555, *p* = 0.0017), the PM (r = −0.5887, *p* = 0.0063), the VCL (r = −0.6580, *p* = 0.0016), the VSL (r = −0.5665, *p* = 0.0092), the VAP (r = −0.6208, *p* = 0.0035), the ALH (r = −0.6270, *p* = 0.0031) and the BCF (r = −0.4516, *p* = 0.0456). In the case of the double-strand breaks in the sperm DNA (neutral comet), no significant correlations (*p* > 0.05) were observed with the sperm motility parameters.

### 3.5. Correlations of Membrane LPO and Sperm DNA Fragmentation with Sperm Functionality Parameters after Thawing

Figure 5 shows the correlations between the membrane LPO and sperm DNA fragmentation (the incidence of global and double-strand DNA breaks) with the sperm functionality parameters of the post-thaw horse sperm. The sperm membrane LPO was negatively correlated with the integrity of the sperm plasma membrane (SYBR14^+^/PI^−^; r = −0.7877, *p* = 0.0001) and positively correlated with the percentage of viable sperm with high intracellular levels of total ROS (DCF^+^/PI^−^; r = 0.6509, *p* = 0.0019), the fluorescence intensity of the total ROS (DCF^+^) of the viable sperm population with high levels of intracellular ROS (r = 0.5260, *p* = 0.0172) and the superoxide fluorescence intensity (E^+^) of the viable sperm population with high intracellular levels of superoxide anion (O_2_^−^) (r = 0.4581, *p* = 0.0422). No correlations (*p* > 0.05) were observed between LPO and the rest of the sperm functionality parameters measured. 

The incidence of global sperm DNA damage (alkaline comet) was negatively correlated with the plasma membrane integrity (SYBR14^+^/PI^−^; r = −0.7243, *p* = 0.0003) and positively correlated with the percentage of viable sperm with high intracellular levels of total ROS (DCF^+^/PI^−^; r = 0.5057, *p* = 0.0229), the fluorescence intensity of total ROS (DCF^+^) of the population of viable sperm with high levels of intracellular ROS (r = 0.6266, *p* = 0.0031) and the superoxide fluorescence intensity (E^+^) of the viable sperm population with high intracellular levels of superoxides (r = 0.6057, *p* = 0.0046). No correlations (*p* > 0.05) were observed between the global sperm DNA damage and the rest of the sperm functionality parameters measured. In the case of double-strand DNA breaks (neutral comet), no significant correlations (*p* > 0.05) were observed with the sperm functionality parameters.

### 3.6. Correlations of the Degree of Lipid Peroxidation and Sperm DNA Fragmentation of Horse Sperm after Thawing with the Levels of Antioxidant Activity (Enzymatic and Non-Enzymatic) and the OSI of SP

Figure 6 shows the correlations between the membrane LPO and sperm DNA fragmentation (the incidence of global and double-strand DNA breaks) of the post-thawing horse sperm with antioxidant activity levels (enzymatic and non-enzymatic) and the OSI of the SP. The sperm membrane LPO was negatively correlated with the levels of the antioxidant enzyme PON1 activity (r = −0.6580, *p* = 0.0016) and the non-enzymatic antioxidant TEAC (r = −0.5843, *p* = 0.0068) and was positively correlated with the OSI of the SP (r = 0.6565, *p* =0.0017). The incidence of global damage to the sperm DNA (alkaline comet) was also negatively correlated with the activity of the antioxidant enzyme PON1 (r = −0.7344, *p* = 0.0002) and the levels of the non-enzymatic antioxidant TEAC (r = −0.6252, *p* = 0.0032) and was positively correlated with the OSI of the SP (r = 0.6419, *p* = 0.0023). Conversely, no significant correlations (*p* > 0.05) were observed between the incidence of double-strand breaks in the sperm DNA (neutral comet), the activity levels of antioxidants (enzymatic and non-enzymatic) and the OSI of SP.

## 4. Discussion

Cryopreservation is regarded as a stressful process for sperm, as it increases ROS production and reduces their quality and survival after thawing [67]. Low and controlled levels of ROS are essential for sperm physiological processes such as motility, capacitation, hyperactivation, acrosome reaction and fertilization [68,69,70]. In spite of this, very high levels of ROS can cause an imbalance with antioxidant capacity and lead to OS, which may trigger LPO, protein damage, DNA fragmentation and apoptosis [71,72,73,74]. In particular, horse sperm have a plasma membrane rich in phospholipids composed of a high proportion of PUFAs [75,76,77]. The presence of unconjugated double bonds in these PUFAs makes these cells especially vulnerable to damage by oxygen free radicals [70,78]. As sperm have limited antioxidant capacity, they are highly dependent on the antioxidant activity of SP, which is removed prior to cryopreservation [29,31,32]. Some studies, nevertheless, suggest that the short time that sperm are in contact with SP before its removal is sufficient for these antioxidants to exert a beneficial effect on sperm cryotolerance [25,29,30,45]. This study showed differences in LPO and the incidence of global DNA fragmentation in frozen–thawed horse sperm between GFEs and PFEs. Additionally, our results demonstrated a positive correlation between LPO and global DNA fragmentation, with increased intracellular levels of ROS in the sperm and OSI measured in the SP. Conversely, LPO and global DNA fragmentation showed a negative correlation with the activity levels of the PON1 and TEAC antioxidants present in the SP.

Lipid peroxidation is the result of an oxidative attack on membrane lipids, mainly phospholipids and cholesterol, changing the permeability and fluidity of the sperm plasmalemma [70,79,80]. As this and other previous studies indicate, LPO compromises the integrity of the plasma membrane, affects sperm function and may also lead to motility impairment and the induction of apoptotic-like changes [9,81]. Whereas the greater LPO observed in the sperm membrane of the PFEs compared to the GFEs observed in the present work was similar to that found in frozen–thawed bovine sperm [82] and frozen–thawed epididymal sperm from buffaloes [83], Gómez-Fernández et al. [84] detected similar LPO and ROS levels in pig sperm when PFEs and GFEs were compared. Such disparities could be attributed to the post-thawing time of the samples used for clustering the GFEs and PFEs (immediately after thawing in horses, cattle and buffaloes; 150 min after thawing in pigs) or to interspecies differences. Differences have been reported with respect to the impact of assisted reproductive technologies (ARTs) on sperm redox status between species [85]. While cryopreservation induces OS in the sperm of horses [86,87,88,89], cattle [90,91,92,93] and cats [94], there is still controversy as to whether it induces OS in the sperm of pigs [67,94,95,96,97,98,99] and dogs [100,101,102]. Interestingly, the association observed between increased membrane LPO and decreased sperm quality after thawing was also found in other studies on horses [62,75,88,103], buffaloes [83,104,105,106], cattle [82], humans [107,108], sheep [109,110] and even birds [111] and fish [112]. In some of them, it was also noted that the increase in LPO was concomitant with an increase in ROS [75,82,83,88,104,105,107,108]. Again, the exception seems to be the pig, where cryopreservation is associated with a loss of sperm quality but not with an increase in LPO and ROS [67].

In addition to its effects on the sperm plasma membrane, LPO can also damage DNA. DNA peroxidation can cause chromatin cross-linking, base shifts and DNA strand breaks [113,114,115]. Several researchers have reported that DNA damage in human sperm occurs together with membrane LPO [116,117,118] and OS [113,115,119,120,121,122]. Our results of DNA fragmentation in horse sperm after freeze-thawing showed an increase in the global incidence of DNA breaks in the PFEs compared to the GFEs, similarly to Hernández et al. [123], who observed greater sperm DNA integrity in GFEs compared to PFEs in frozen–thawed pig semen, and comparable to those documented in frozen–thawed sperm from bulls with varying fertility levels, where the group with lower fertility displayed increased DNA fragmentation and ROS levels, along with reduced sperm quality parameters [124]. Furthermore, our findings of high DNA fragmentation and its correlation with decreased sperm quality and increased ROS were consistent with previous findings in frozen–thawed sperm from horses [125], cattle [92,126], pigs (ROS was not analyzed) [127], sheep [128] and humans [129]. Regarding the incidence of DSBs, no differences between the GFEs and PFEs were found in the present study. Furthermore, the incidence of DSBs was not correlated with sperm quality parameters and ROS, which is consistent with the previous report by Ribas-Maynou et al. [129], who did not observe any effect of cryopreservation on double-strand DNA breaks. One can thus posit that the differences in sperm DNA fragmentation between GFEs and PFEs after thawing are due to SSBs rather than DSBs and are related to the levels of ROS leading to oxidative stress, which, as in the case of LPO, vary between ejaculates of good and poor cryotolerance.

It has been suggested that variability in sperm cryotolerance between stallions and/or ejaculates may be partly related to the composition of their SP [25,29,30,39,40,41]. The processing of semen for cryopreservation involves the removal of SP and its natural ROS controllers/scavengers, such as antioxidants, ergothioneine, glyceryl phosphocholine, etc., and reduces—but does not eliminate—the ability of sperm to counteract oxidative stress [75]. For instance, Ball et al. [130] demonstrated that the removal of SP from equine semen did not eliminate sperm-associated catalase activity. It should be noted that when semen is processed for storage, the removal of SP is never complete [75]. Previous research showed that the levels of some SP antioxidants were directly related to sperm’s resilience to cryopreservation [25,30,45,47]. The results obtained in this study would agree with this hypothesis, as detailed below.

Regarding the enzymatic antioxidants, no correlation was found between the activity levels of the antioxidants SOD, CAT and GPX with LPO and DNA fragmentation, which, in the case of LPO’s relationship with CAT and GPX, was similar to the results reported by Ortega-Ferrusola et al. [62] in frozen–thawed horse semen. However, in this study, the activity levels of PON1 showed a negative relationship with LPO and global DNA fragmentation in the frozen–thawed sperm. Whereas this relationship is comparable to that published in boar by Li et al. [47], it is reported here for the first time in equids and could confirm the importance of the activity levels of this enzyme in the SP in sperm cryotolerance. In the case of DNA fragmentation, although no relationship has previously been reported in frozen sperm, our data are similar to those of porcine semen stored at 17 °C for 24 h [131], where a positive correlation between SP PON1 activity levels and a higher integrity of sperm DNA in the sperm-rich fraction was observed. PON1 is an enzyme located in the extracellular space and associated with high-density lipoproteins (HDLs). It has anti-inflammatory properties and prevents the oxidation of low- and high-density lipoproteins [132,133] by binding to membrane cholesterol and hydrolyzing specific lipid peroxides such as cholesteryl esters and oxidized phospholipids [46,134,135,136]. Since oxidative stress is a major cause of sperm DNA fragmentation [137,138], the negative correlation between PON1 activity and DNA integrity may be due to the protective role of PON1 against oxidative injury induced by excessive ROS production. This, together with the negative correlation of the SP PON1 with LPO and the results obtained in previous studies in frozen–thawed horse sperm, where the importance of PON1 levels in sperm cryotolerance could be observed [29], suggests that SP PON1 may be an essential ROS scavenger in horse semen.

Regarding the relationship of non-enzymatic antioxidants, our results, showing a negative correlation of the TEAC with LPO and DNA fragmentation—the first time reported in equines—are similar to findings by Alyethodi et al. [82] in bovine semen. They observed a lower TEAC and higher LPO and ROS in poor-freezability ejaculates compared to good-freezability ejaculates. Our data also agree in part with those reported by Gürler et al. [139], who observed a negative correlation between TEAC levels and LPO in frozen–thawed bovine sperm. However, unlike our study, they did not find a negative correlation between the TEAC and sperm DNA fragmentation, which could be attributed to the low number of samples analyzed for this parameter (four ejaculates). Li et al. [47] and Barranco et al. [140] also found that higher TEAC levels in the sperm-rich fraction of porcine SP were associated with lower levels of LPO and ROS in frozen–thawed sperm. Indeed, horse SP has been identified to modulate LPO in frozen–thawed sperm [103], although it was not analyzed which components of SP could be attributed to such beneficial effects on LPO. The redox state of horse SP is crucial for sperm cryotolerance [29] and influences DNA integrity/damage (SSBs and DSBs), as indicated by biomarkers like 8-hydroxyguanosine [141].

Overall, the correlations observed in this study between SP OSI with LPO, global DNA damage and sperm quality parameters post-thawing suggest the significance of a proper balance between oxidants and antioxidants in SP concerning LPO and DNA fragmentation in horse sperm. The present results confirm the importance of this balance in the capacity of horse sperm to withstand freezing and thawing processes.

Finally, it is important to mention that, while efforts have been made to identify markers of sperm freezability in different species, including horses, the results of this study indicate that the activity of PON1, the TEAC and the OSI in horse SP are related to sperm cryotolerance and could be used as potential biomarkers of freezability. However, it is also necessary to take into account some limitations of this study, such as assessing whether the activity levels of these antioxidants in SP are related to the fertility of these ejaculates and/or if the addition of these antioxidants could help reduce LPO and DNA damage, thereby rescuing sperm function and survival during preservation.

## 5. Conclusions

In conclusion, this study found differences in the membrane LPO and the incidence of global DNA breaks when the frozen–thawed sperm of horse ejaculates of different cryotolerances (GFEs and PFEs) were compared. Furthermore, the LPO and the incidence of global DNA breaks in frozen–thawed sperm were found to be positively correlated with ROS levels and the SP OSI. Moreover, the post-thaw sperm quality was seen to be negatively correlated with the levels of antioxidant activity (PON1 and TEAC) present in the SP. It could be hypothesized that the differences observed in the LPO and DNA fragmentation of the frozen–thawed sperm from the different stallions/ejaculates could be influenced by the antioxidant activity of their SPs (PON1 and TEAC). The balance between the oxidant and antioxidant components present in SP would thus help to control ROS levels in sperm and prevent the adverse effects of oxidative stress on sperm cells. Further research addressing the relationship of these antioxidants and the SP OSI with LPO and sperm DNA fragmentation in horses and other species is warranted. This could allow us to (1) select those ejaculates that are likely to withstand cryopreservation best and (2) identify which semen samples may require antioxidant supplements for storage.

## Figures and Tables

**Figure 1 antioxidants-13-00322-f001:**
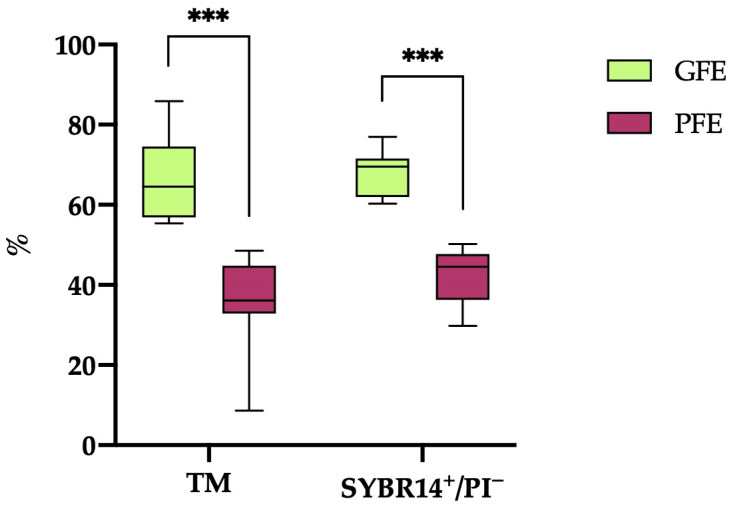
The box-and-whisker plot displays the percentages of total motile (TM) and viable (SYBR14^+^/PI^−^) sperm following thawing in horse ejaculates categorized as having good freezability (GFEs, n = 12; lime) or poor freezability (PFEs, n = 8; burgundy). The boxes represent the interquartile range (25th to 75th percentiles), the whiskers extend to the 5th and 95th percentiles and the median is denoted by a line. (***) indicates statistical significance at *p* ≤ 0.001.

**Figure 2 antioxidants-13-00322-f002:**
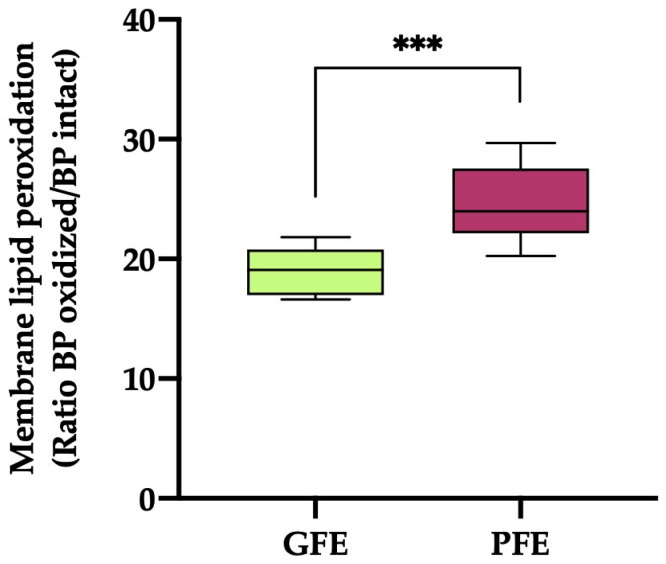
The box-and-whisker plot shows sperm membrane lipid peroxidation (ratio of BODIPY^581/591^-C11 (BP) oxidized/BP intact) in horse ejaculates classified as having good (GFEs, n = 12; lime) or poor freezability (PFEs, n = 8; burgundy). The boxes enclose the 25th and 75th percentiles, the whiskers extend to the 5th and 95th percentiles and the line indicates the median. (***) *p* ≤ 0.001.

**Figure 3 antioxidants-13-00322-f003:**
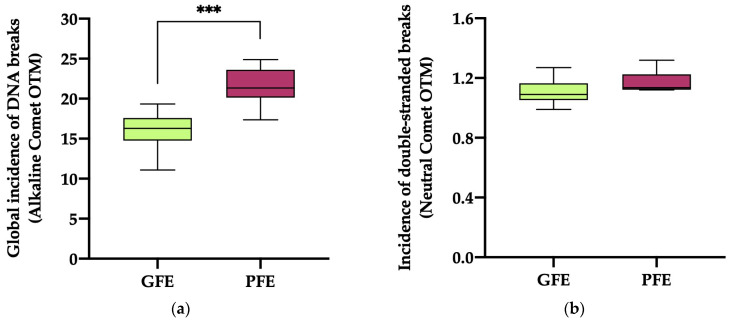
The box-and-whisker plot displays sperm DNA fragmentation in horse ejaculates classified as having good (GFEs, n = 12; lime) or poor freezability (PFEs, n = 8; burgundy), measured as the global incidence of DNA damage (based on OTM) ((**a**); alkaline comet) and the incidence of double-strand DNA breaks (based on OTM) ((**b**); neutral comet). The boxes enclose the 25th and 75th percentiles, the whiskers extend to the 5th and 95th percentiles and the line indicates the median. (***) *p* ≤ 0.001.

**Figure 4 antioxidants-13-00322-f004:**
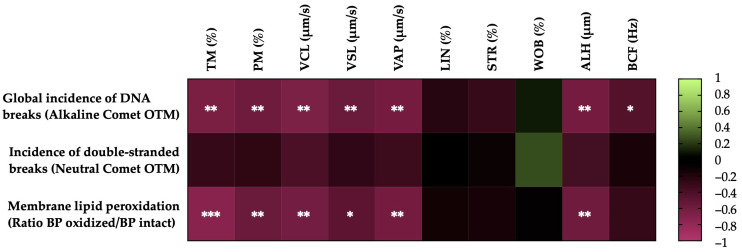
Heat map showing the correlations of membrane lipid peroxidation and sperm DNA fragmentation (the incidence of global and double-strand DNA damage) with horse sperm motility parameters after thawing (n = 20) (total motility, TM; progressive motility, PM; curvilinear velocity, VCL; straight line velocity, VSL; average path velocity, VAP; linearity coefficient, LIN; straightness coefficient, STR; wobble coefficient, WOB; amplitude of lateral head displacement, ALH; and beat-cross frequency, BCF). The colors on the scale (1 to −1) indicate whether the correlation is positive (lime) or negative (burgundy). (*) *p* ≤ 0.05; (**) *p* ≤ 0.01; (***) *p* ≤ 0.001.

**Figure 5 antioxidants-13-00322-f005:**
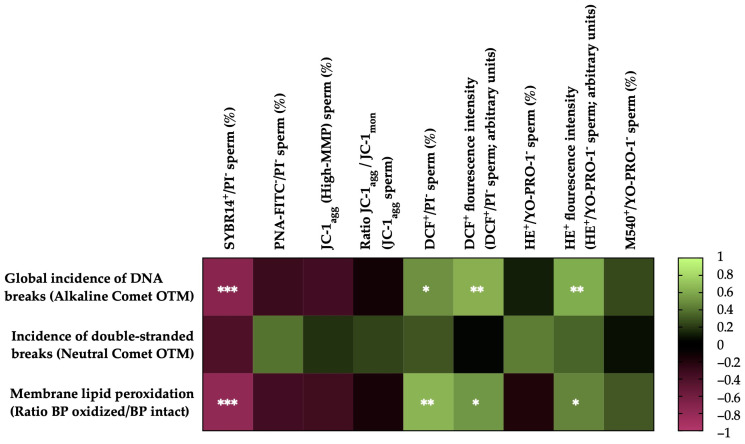
Heat map showing correlations of membrane lipid peroxidation and sperm DNA fragmentation (the incidence of global and double-strand DNA damage) with sperm functionality parameters recorded after thawing (n = 20) (plasma membrane integrity, SYBR14^+^/PI^−^; acrosome membrane integrity, PNA-FITC^−^/PI^−^; mitochondrial membrane potential, MMP, JC-1_agg_ and ratio between JC-1 aggregates (JC-1_agg_) and JC-1 monomers (JC-1_mon_) for the sperm population with high mitochondrial membrane potential; intracellular ROS levels, DCF^+^/PI^−^ and DCF^+^ fluorescence intensity; intracellular superoxide levels, E^+^/YO-PRO-1^−^ and HE^+^ fluorescence intensity; and plasma membrane lipid disorder, M540^+^/YO-PRO-1^−^). The colors on the scale (1 to −1) indicate whether the correlation is positive (lime) or negative (burgundy). (*) *p* ≤ 0.05; (**) *p* ≤ 0.01; (***) *p* ≤ 0.001.

**Figure 6 antioxidants-13-00322-f006:**
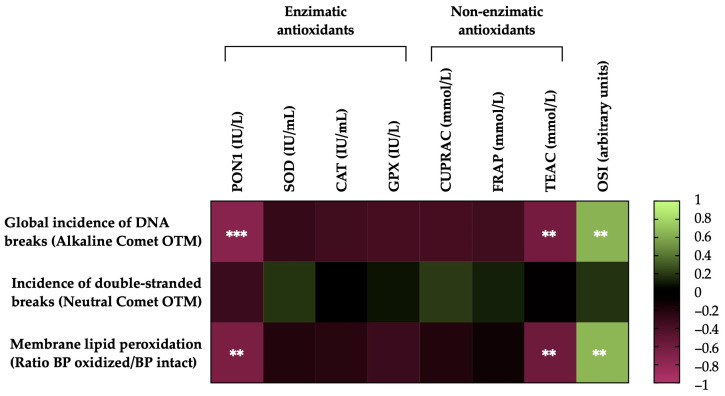
Heat map showing correlations of membrane lipid peroxidation and sperm DNA fragmentation (the incidence of global and double-strand DNA damage) after thawing (n = 20), with the activity levels of enzymatic (paraoxonase type 1, PON1; superoxide dismutase, SOD; catalase, CAT; and glutathione peroxidase, GPX) and non-enzymatic antioxidants (measured in terms of cupric-reducing antioxidant capacity, CUPRAC; ferric-reducing ability of plasma, FRAP; and Trolox-equivalent antioxidant capacity, TEAC) and the oxidative stress index (OSI) of seminal plasma (SP) in horse ejaculates. The colors on the scale (1 to −1) indicate whether the correlation is positive (lime) or negative (burgundy). (**) *p* ≤ 0.01; (***) *p* ≤ 0.001.

## Data Availability

All data are contained within the article.

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
