# Peer review of "Impact of Seminal Plasma Antioxidants on DNA Fragmentation and Lipid Peroxidation of Frozen–Thawed Horse Sperm"

_antioxidants, 2024, doi:10.3390/antiox13030322_

Round 1

Reviewer 1 Report (Previous Reviewer 2)

It is considered that with the modifications made to the original manuscript, and with the explanations sent by the authors, everything is ready for publication.

They have done a great job.

Reviewer 2 Report (Previous Reviewer 1)

The Authors have addressed all my concerns and I have no further comments. As far as I am concerned, the manuscript is now acceptable to be published.

The Authors have addressed all my concerns and I have no further comments. As far as I am concerned, the manuscript is now acceptable to be published.

This manuscript is a resubmission of an earlier submission. The following is a list of the peer review reports and author responses from that submission.

Round 1

Reviewer 1 Report

Comments and Suggestions for Authors

In this study, Catalán et al. investigated if the antioxidant components of horse seminal plasma are related to DNA fragmentation and membrane lipid peroxidation of frozen-thawed horse sperm.
The study essentially confirms the importance of a proper balance between oxidants and antioxidants in seminal plasma in the ability of horse sperm to withstand freezing and thawing.
The Authors should therefore better emphasize what is new in this study compared to what already exists in the literature and what implications this study has for animal reproduction.
The limitations of this study should also be taken into account.
Based on this observation, the introduction and the discussion section should be rewritten.
In particular, the introduction must be more focused on horse seminal plasma and spermatozoa and highlight the limitations associated with cryopreservation techniques, but also the advantages of these techniques. In this article, only lines 47-52 refer to horse sperm, and the references cited refer to the seminal fluid of other animal species. In addition, lines 89-95 describe the purpose of the study, but the stated purpose does not appear to be innovative, but rather to confirm aspects already known in the literature.
At the same time, the discussion must focus on the new results obtained from this study and their implications. Once again lines 635-637 seem like just a confirmation of previous studies.
A paragraph with the limitations of the study should also be included, in order to generalize the conclusions and implications in the field of animal assisted reproduction.

Minor:
Line 40: please define that SP is seminal plasma.
Line 423:  MT or TM?

Author Response

Reviewer #1

General comment: In this study, Catalán et al. investigated if the antioxidant components of horse seminal plasma are related to DNA fragmentation and membrane lipid peroxidation of frozen-thawed horse sperm.

The study essentially confirms the importance of a proper balance between oxidants and antioxidants in seminal plasma in the ability of horse sperm to withstand freezing and thawing. The Authors should therefore better emphasize what is new in this study compared to what already exists in the literature and what implications this study has for animal reproduction. The limitations of this study should also be taken into account. Based on this observation, the introduction and the discussion section should be rewritten.

Answer: We are sincerely grateful for both your constructive feedback and critical insights, which have been instrumental in refining our research and have significantly contributed to the enhancement of our manuscript. We have implemented the recommended revisions and addressed the queries raised by the reviewer.

Specific Comment 1: In particular, the introduction must be more focused on horse seminal plasma and spermatozoa and highlight the limitations associated with cryopreservation techniques, but also the advantages of these techniques. In this article, only lines 47-52 refer to horse sperm, and the references cited refer to the seminal fluid of other animal species. In addition, lines 89-95 describe the purpose of the study, but the stated purpose does not appear to be innovative, but rather to confirm aspects already known in the literature.

Author Answer: Thank you very much for your suggestion. In accordance with the reviewer's recommendation, the introduction has been modified to focus more on equine seminal plasma and spermatozoa. New paragraphs and citations related to the studied species have been incorporated, emphasizing both the advantages and limitations associated with cryopreservation techniques. Additionally, the introduction highlights the importance and novelty of the study, aiming to present it not merely as a confirmation of already known aspects.

Specific Comment 2: At the same time, the discussion must focus on the new results obtained from this study and their implications. Once again lines 635-637 seem like just a confirmation of previous studies.

Author Answer: Thank you very much for your suggestion. In accordance with the reviewer's recommendation, the discussion has been modified to focus on the new results and discuss their potential implications. This aims to present the study not as a mere confirmation of previous findings but as a contribution with its own insights and significance.

Specific comment 3: A paragraph with the limitations of the study should also be included, in order to generalize the conclusions and implications in the field of animal assisted reproduction.

Author Answer: Thank you very much for your suggestion. In line with the reviewer's advice, the study limitations have been included in the discussion. Possible implications of the results of this study in the field of assisted reproduction in animals have also been incorporated.

Minor comment 1: Line 40: please define that SP is seminal plasma.

Author Answer: Thank you very much for your comment. This was revised according to the reviewer's suggestion, and the definition that "SP" refers to seminal plasma was provided earlier in the text, specifically in L. 31.

Minor comment 2: Line 423: MT or TM?

Author Answer: Thank you very much for your comment. This typo mistake was corrected in the manuscript.

Reviewer 2 Report

Comments and Suggestions for Authors

This is an interesting paper, and the way it has presented its complex methodology is valued very positively. As well as the presentation of the results, which in this type of study is not easy, due to the number of parameters that are obtained.   Questions:   - Why choose the Kenney extender? - "sperm viability and morphology were examined through eosin-nigrosine staining"(125). But the cell morphology has not been presented as a result. - In the GFE or PFE classification, is there fertility data for these horses in artificial insemination of mares?   The conclusion and hypothesis advanced are interesting, and more studies will be needed to corroborate these results obtained in this paper.  

Author Response

Reviewer #2

General comments:

This is an interesting paper, and the way it has presented its complex methodology is valued very positively. As well as the presentation of the results, which in this type of study is not easy, due to the number of parameters that are obtained.

Author Answer: We sincerely appreciate your valuable feedback and the comprehensive review of our manuscript. We have implemented the recommended revisions and addressed the queries raised by the reviewer.

Specific Comment 1: Why choose the Kenney extender?

Author Answer: Thank you very much for your question. Because fresh semen, once the seminal analysis is performed, would be immediately centrifuged, and the supernatant (which contains the extender) would be removed, we chose to use a basic and cost-effective extender such as the Kenney extender. This extender has been widely used and tested as an effective medium for preserving the motility, viability, and fertility of horse semen. For the same reason, this Kenney extender was also used to dilute the thawed semen, as the sole objective was to dilute the semen to obtain a sperm concentration suitable for motility analysis through the CASA system.

Specific Comment 2: “sperm viability and morphology were examined through eosin-nigrosine staining"(125). But the cell morphology has not been presented as a result.

Author Answer: Thank you very much for your comment. As the reviewer mentions, after the collection, a routine analysis of the sperm quality of diluted fresh semen was conducted, evaluating parameters such as viability and morphology through eosin-nigrosine staining. However, the data on sperm morphology were not presented in this study. This is because this analysis (viability and morphology of diluted fresh semen by eosin-nigrosine), as well as sperm motility analysis (evaluated by the CASA system), was used to determine whether ejaculates met the minimum standard to be included in the study (>65% total motility, >65% viable sperm, and >70% morphologically normal sperm). This was done to ensure that the classification of ejaculates as good or poor freezers was influenced by the impact of the semen freezing-thawing procedures on sperm and not by the initial sperm quality parameters of each ejaculate. After semen thawing, viability analysis was performed using flow cytometry, and sperm morphology was not evaluated. Therefore, this parameter is not included in the presented results.

Specific Comment 3: In the GFE or PFE classification, is there fertility data for these horses in artificial insemination of mares?

Author Answer: Thank you very much for your question. The animals from which the ejaculates used in this study originate are owned by private breeders who turn to the Equine Reproduction Service at the Autonomous University of Barcelona for semen cryopreservation. Therefore, while there are fertility data available for some stallions used, we do not have fertility information for all animals. A significant portion of them visited this center exclusively for the cryopreservation of their semen, and after this process, the semen is sent to various locations around the world.

Specific Comment 4: The conclusion and hypothesis advanced are interesting, and more studies will be needed to corroborate these results obtained in this paper. 

Author Answer: Thank you very much for your comment. A paragraph in the conclusions section has been added addressing the reviewer's comment.

Round 2

Reviewer 1 Report

Comments and Suggestions for Authors

The Authors have addressed all my concerns and I have no further comments. 

As far as I am concerned, the manuscript is now acceptable to be published.